# Hierarchical Contrastive Reinforcement Learning: learn representations more suitable for RL environments

## Abstract

Goal-conditioned reinforcement learning with sparse reward holds significant importance for real-world environments. Many researchers have tried to improve performance for this problem on various simulation benchmarks. In a simulation environment, the goals are represented either as a full future state or as a subset of dimensions within the state vector. However, this setting is just for simplification and does not reflect the real-world environment. Consequently, some methods that achieve good performance in simulation often face limitations when applied to real-world environments. Based on previous contrastive reinforcement learning algorithms, this work presents a new approach **Hierarchical Contrastive Reinforcement Learning** that allows the goal representation and the state representation to be independent. Our method designs a hierarchical structure to drive the agent to first understand the relationship between action and state, and then learn the relationship between state and goal. The results of experiments show that HCRL provides faster convergence and higher success rate in goal-conditioned reinforcement learning environments with sparse reward, without introducing any additional assumptions or constraints. We further conduct ablation studies and additional evaluations to validate our method. Anonymous code: https://anonymous.4open.science/r/HCRL-6E88.

## 1 Introduction

In the real world, tasks are often linked to goals. Correct actions are determined by both the environment and the goals, without complex rewards. This is precisely the significance of goal-conditioned reinforcement learning with sparse rewards. Many researchers have proposed various solutions from different perspectives. Early methods focused on improving sampling efficiency to facilitate goal attainment (Andrychowicz et al., 2017; Chane-Sane et al., 2021; Ding et al., 2019). However, these approaches can introduce bias and often perform poorly in complex environments. Subsequently, researchers found that learning a Q function independent of the reward and a representation of the state is an effective approach (Finn et al., 2016; Guo et al., 2018; Lange & Riedmiller, 2010). Some of them separated learning representations from training RL agents, but this makes it difficult to evaluate the quality of representations (Achiam et al., 2019; Laskin et al., 2022). Others directly used the similarity or distance between representations learned via contrastive learning as an indicator of the action taken by the agent (Emmons et al., 2021; Eysenbach et al., 2020b; 2022). These methods greatly improve the data efficiency and convergence of goal-conditioned RL with sparse rewards, and provide a seemingly elegant paradigm through contrastive learning: it provide a dense and reasonable reward function without complex reward shaping.

However, these methods such as CRL (Eysenbach et al., 2022) face significant limitations when applied to real-world tasks, particularly in navigation and robotics. This is because the goal representations used in simulation environments do not fully align with real-world settings. In real-world, the goal might be a specific position of an object in a task. More commonly, the goal may be a language instruction rather than a specific position, in which there might be a set of acceptable goals. By contrast, in simulation, goals are often represented as a full future state or as a subset of dimensions within the state vector. There exists a substantial gap between reality and simulation.

We briefly discuss why CRL doesn't perform well when the goal and the state are defined in separate representation spaces. One of the core designs of CRL is to improve the data efficiency of contrastive learning by sampling future achieved goals along the trajectory. In a simulation environment, the representation of achieved goals on a trajectory is diverse, because the achieved goals are just the future states. However, if the goal is a language instruction, only the last time step in a trajectory will be "achieved," while the other time steps will be "not achieved". This could reduce the data efficiency of CRL and other similar methods, making them difficult to converge. Furthermore, if the goal can be represented by position or other precise metrics in real-world, it is similar to representing the goal using several dimensions of state in a simulation environment. And CRL still has room for improvement in such an environment setting, especially in robotics. The focus remains on the diversity of the goal representation along a trajectory. For example, in a task where a robotic arm moves an object to a specific position, the goal is represented by the object's position. The goal remains constant until the robotic arm moves the object, which also reduces the performance of CRL.

We try to design a new structure to improve these problems. Before introducing the approach presented in this paper, we emphasize that although CRL has the limitations discussed above, it provides a broadly applicable paradigm for obtaining dense rewards by using distances between learned representations, without requiring complex reward shaping. This paradigm is both insightful and meaningful for practical reinforcement learning applications. Therefore, our goal is not to replace it with an entirely different algorithm, but rather to build upon and improve it.

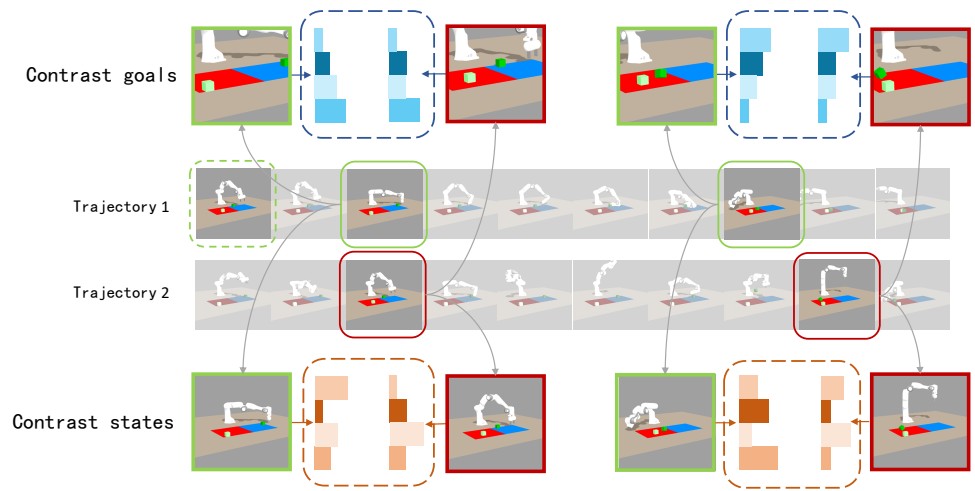

Figure 1: The diversity of samples critically influences the effectiveness of contrastive learning. While sampled goals may be similar or even identical, sampled states could exhibit sufficient diversity to ensure meaningful representation learning.

Our algorithm, HCRL, overcomes the above challenges by introducing intermediate representations via a hierarchical structure. In both scenarios, the core issue arises from the fact that sampling achieved goals from trajectories lacks diversity, which degrades the effectiveness of contrastive learning. To mitigate this, we introduce an intermediate feature space associated with the state. In the reinforcement learning setting, the agent executes actions at each timestep and induce state transitions, which means the states are diverse. Specifically, we first compare the state-action pair $(s, a)$ with a future state $s'$ along the same trajectory, mapping it into a new feature space. Next, we compare the vectors of the intermediate representations in this space with the goal, producing the final representation. This hierarchical structure implies a two-stage learning process: first, the agent learns how to reach a specific state through its actions; second, it learns to assess whether a given state is proximal to the final goal. And the first comparison in HCRL overcomes the challenge of insufficient goal diversity in sampling.

But there are two issues that remain. First, why is the second comparison not affected by goal similarity along the same trajectory? Second, why not directly employ a pre-trained model, such

as CLIP, to compare states and goals? Regarding the first issue, it is indeed still influenced by goal similarity. However, the key distinction is that the second comparison involves a simpler learning objective: determining whether a state is close to the goal. This is substantially easier than learning which action to take to reach the goal, thereby reducing the reliance on goal diversity. Concerning the second issue, the feature space of a pre-trained model may exhibit low correlation with the intermediate representations obtained from the first comparison. In HCRL, the two comparisons are tightly integrated, ensuring coherence between intermediate and final representations.

In this paper, we make the following main contributions:

1) We analyze the limitations of previous contrastive reinforcement learning approaches and provide a new perspective on how goal-conditioned reinforcement learning can be applied in real-world settings.

2) We provided a new algorithm structure to address these challenges and introduced several designs to the algorithm to improve its performance. And we also provide error analysis and convergence analysis.

3) We conduct comparative experiments to show that our method performs better than prior work in a variety of environments. We also conducted ablation experiments and other experiments to validate the effectiveness.

## 2 RELATED WORK

### 2.1 GOAL-CONDITIONED RL

GCRL is a special case of general RL settings, in which each task is represented by a specific goal, and the agent can know whether it has achieved this goal (Ghosh et al., 2018; Nair et al., 2018; Schaul et al., 2015). Prior work has approached this problem using different methods. Temporal difference learning aims to improve convergence under sparse rewards, often accompanied by the idea of increasing sample efficiency (Andrychowicz et al., 2017; Eysenbach et al., 2020b; Lin et al., 2019; Rudner et al., 2021; Eysenbach et al., 2018). Conditional imitation learning can effectively utilize expert demonstration (Ding et al., 2019; Ghosh et al., 2019; Lynch et al., 2020; Savinov et al., 2018). Model-based methods can efficiently plan and conduct virtual exploration under target conditions by learning the dynamics model of the environment (Dosovitskiy & Koltun, 2016; Schmeckpeper et al., 2020). And some methods for automatic sampling and exploration of targets (Du et al., 2021; Florensa et al., 2018; Mendonca et al., 2021; Pong et al., 2019; Zhao et al., 2019). In addition, some methods regard goal-conditioned RL as a data-driven problem, rather than a reward-maximization problem (Blier et al., 2021; Chane-Sane et al., 2021; Eysenbach et al., 2020b). Hindsight relabeling is a general tool of these methods (Andrychowicz et al., 2017; Eysenbach et al., 2020a; Li et al., 2020). And some goal-conditioned methods train a value function that quantifies the similarity between two states (Nair et al., 2018; Wang et al., 2024). There are also some elegant and effective methods, such as QRL (Wang et al., 2023).

### 2.2 CONTRASTIVE LEARNING

Contrastive learning is a representation learning method that learns discriminative representations by narrowing the feature distances between positive pairs and widening the feature distances between negative pairs (Chen et al., 2020; Hjelm et al., 2018; Hoffer & Ailon, 2015; Levy & Goldberg, 2014; Mikolov et al., 2013; Mnih & Teh, 2012; Nowozin et al., 2016; Oord et al., 2018; Schroff et al., 2015; Sohn, 2016; Tian et al., 2020; Weinberger & Saul, 2009; Wu et al., 2018). Initially, researchers used labeled datasets to construct positive and negative pairs to extract representations. Subsequently,researchers have emerged to adapt contrastive learning for self-supervised learning. Common methods for data augmentation include cameras with different viewpoints (Sermanet et al., 2018; Tian et al., 2020), and samples with similar temporal proximity in time series data (Anand et al., 2019; Oord et al., 2018; Sermanet et al., 2018; Stooke et al., 2021a). Contrastive learning is widely used in fields such as NLP and CV due to good convergence and strong generalization.

## 2.3 RL with Representations

Learning representations in RL is an effective approach to improve accuracy and enhance generalization. Learning low-dimensional representations of high-dimensional environments can effectively reduce the complexity of reinforcement learning algorithms. But prior works have found it challenging to learn good representations (Finn et al., 2016; Guo et al., 2018; Lange & Riedmiller, 2010; Laskin et al., 2020; Nachum et al., 2018; Nair et al., 2018; Liang et al., 2015). The learning objectives of some methods is to reconstruct the input state (Finn et al., 2016; Ha & Schmidhuber, 2018; Hafner et al., 2019a;b; Lange & Riedmiller, 2010; Nair et al., 2018; Qiu et al., 2022; Nasiriany et al., 2019; Rakelly et al., 2021). And others use contrastive representation methods. Representations are generally used to acquire reward functions (Brown et al., 2019; Christiano et al., 2017; Fu et al., 2018; Kalashnikov et al., 2021; Konyushkova et al., 2020; Nair et al., 2022; Xie et al., 2018; Xu & Denil, 2021; Zolna et al., 2021) or used in imitation learning (Fu et al., 2017; Ho & Ermon, 2016). Totally, these methods employ separate objectives for representation learning and RL (Stooke et al., 2021b; Zhang et al., 2020; 2022).

If we could directly learn representation irrelevant to reward in RL training, we will have an effective and universal RL algorithm. An interesting approach is to learn a value function that captures the similarity between two states (Eysenbach et al., 2020b; Kaelbling, 1993; Nair et al., 2018; Venkattaramanujam et al., 2019). CRL (Eysenbach et al., 2022) performs best in these methods. It provides a novel method that the distance functions are structurally regarded as a Q-function and are used directly to take actions. This approach is valuable for goal-conditioned RL and we conducted further research, and exploration based on it.

# 3 Preliminaries

## 3.1 Goal-Conditioned Reinforcement Learning

The goal-conditioned RL problem is defined by a controlled Markov process (CMP) $\mathcal{M} =< \mathcal{S}, \mathcal{A}, \mathcal{G}, p, p_0, r >$. At time $t$, the agent could observe a state $s_t \in \mathcal{S}$, and select a suitable action $a_t \in \mathcal{A}$. The environment has an initial state $p_0$, and could give a next state $s_{t+1}$ by distribution $p(s_{t+1}|s_t, a_t)$. If the agent could achieve the goal $g \in \mathcal{G}$, this trajectory is considered successful. According to the different reward functions $r$, goal-conditioned RL problems are divided into non-sparse, which reward function is similar to normal RL problems, and sparse, which reward function is generally $r_g = 1 \ if \ s_t = f(g) \ else \ 0$. The function $f$ is used to judge whether the goal is achieved. This sparse reward function could be expressed as:

$$r(s_t, a_t, g) = (1 - \gamma) \, p(s_{t+1} = f(g)|s_t, a_t) \tag{1}$$

The goal-conditioned RL algorithms try to find an optimal policy $\pi(a|s, g)$. Denote $p_g(s_g)$ as the distribution of acceptable termination states $s_g$ when the objective is $g$, and denote $\pi(\tau|s_g)$ as the probability of sampling an infinite-length trajectory $\tau = (s0, a0, s1, a1, ...)$. The objective to be optimized is in Eq. (2) and the Q-function is in Eq. (3):

$$\max_{\pi} \mathbb{E}_{p_g(s_g),\pi(\tau|s_g)} \left[ \sum_{t=0}^{\infty} \gamma^t r\left(s_t, a_t, g\right) \right] \tag{2}$$

$$Q^{\pi}(s, a, g) = \mathbb{E}_{\pi(\tau|s_g)} \left[ \sum_{t'=t}^{\infty} \gamma^{t'-t} r\left(s_{t'}, a_{t'}, g\right) \Big| \ s_t = s, a_t = a \right] \tag{3}$$

The actor-critic architecture is a widely used paradigm in RL that combines the strengths of both value-based and policy-based methods. Specifically, the actor refers to a parameterized policy $\pi(a|s, g)$ that selects actions given the current state, while the critic denotes a value function, typically represented by the action-value function $Q_\phi(s, a, g)$ or the state-value function $V_\phi(s)$. In the goal-conditioned RL algorithms, the loss functions of the actor and the critic are generally:

$$\mathcal{L}_Q(\phi) = \mathbb{E}_{(s,a,r,s',g)} \left[ \left( Q_\phi(s, a, g) - r(s, a, g) - \gamma Q_{\phi^-}(s', \pi_\theta(a|s', g), g) \right)^2 \right] \tag{4}$$

$$\mathcal{L}_\pi(\theta) = -\mathbb{E}_{s,g} \left[ Q_\phi\left(s, \pi_\theta(a|s, g), g\right) \right] \tag{5}$$

### 3.2 Contrastive Reinforcement Learning

CRL uses contrastive learning to evaluate the Q-function in the critic-actor method. To achieve this, the critic of contrastive reinforcement learning usually consists of the following parts: a) the encoder of thestate-action pair $\psi_1(s, a)$ and the encoder of the goal $\psi_2(g)$. b) an energy function $f(s, a, g)$, which is used to measure the similarity or distance of $\psi_1(s, a)$ and $\psi_2(g)$. c) a contrastive loss function, which is used to learn representations by bringing positive sample pairs closer and pushing negative sample pairs further away. Historical trajectories are collected in a data batch $B$. The objective of the CRL critic could be expressed as follows.

$$\mathcal{L}(\psi_1, \psi_2) = \mathbb{E}_{\mathcal{B}} \left[ - \sum_{i=1}^{|\mathcal{B}|} \log \left( \frac{e^{f(\psi_1(s_i, a_i), \psi_2(g_i))}}{\sum_{j=1}^{K} e^{f(\psi_1(s_i, a_i), \psi_2(g_j))}} \right) \right] \tag{6}$$

where $g_i, g_j$ means goals sampled from different trajectories $i, j$. After convergence, the energy function could represent the Q-function.

The actor in CRL tries to find a policy that can give the optimal action $a_t$ under the state $s_t$ when the goal is $g$. The optimal action means that it maximizes the Q-value and minimizes the distance to the goal. We denote $p(s, a)$ as the distribution of state-action pairs and denote $p(g|s, a)$ as the distribution of the goal in a specific trajectory. One kind of actor loss could be expressed as:

$$\mathcal{L}(\theta) = -\mathbb{E}_{p(s,a)p(g|s,a)\pi_\theta(a'|s,g)} \left[ f_{\psi_1, \psi_2} \left( s, a', g \right) \right] \tag{7}$$

## 4 Hierarchical Contrastive Reinforcement Learning

We introduce the structure of our method hierarchical contrastive reinforcement learning, which still follows the Actor-critic architecture. Our work primarily concerns the design of critic.

### 4.1 The Hierarchical Structure of Critic

We designed a hierarchical contrastive learning structure to introduce an intermediate representation space. First, we sample a state–action pair $(s, a)$ from a trajectory at a given timestep and select a future state $s'$ from the same trajectory, which we map into an intermediate representation space denoted $\psi$. Secondly, we sample a goal $g$ from a trajectory at a given timestep and select a past state–action pair $(s, a)$ from the same trajectory, mapping $g$ and $\psi(s, a)$ into the target representation space.

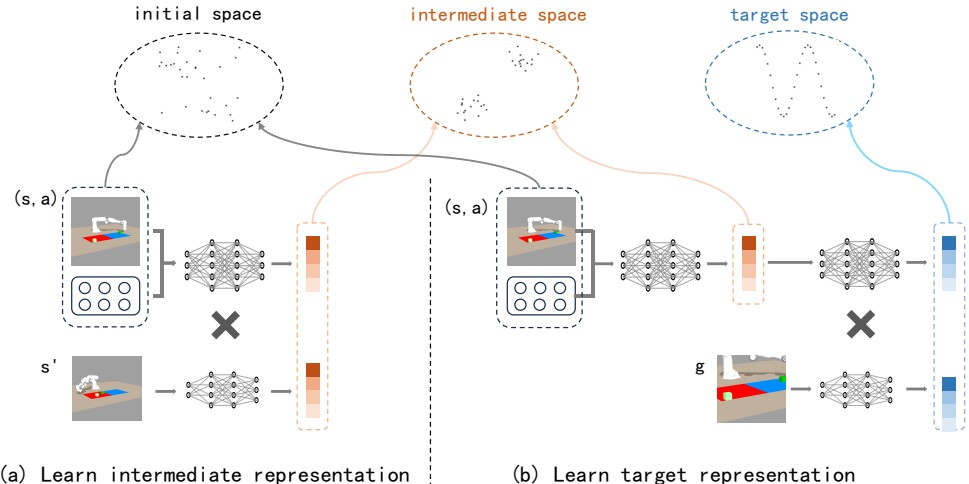

(a) Learn intermediate representation  (b) Learn target representation

Figure 2: First, encode $(s, a)$ and $s'$ into an intermediate space. And then, encode the representation of $(s, a)$ in the intermediate space and $g$ into the target representation.

In this way, we divide the process of learning how to select an action based on a specific goal into two parts: first, learning what future state the action will lead to; and then learning what state is closer to the goal. Any environment naturally provides diverse states, enabling contrastive learning to converge effectively in the first stage, which is determined by the reinforcement learning setup. Although the goal in some environments may be identical or similar, the learning object in the second stage is comparatively simple, making successful learning still attainable.

## 4.2 IMPLEMENTATION DETAILS

In this section, we first give the final loss function and then discuss some details about it.

$$\mathcal{L}(\psi_1, \psi_2) = -\sum_{i=1}^{|\mathcal{B}|} \log\left(\frac{e^{f\left(\psi_1(s_i, a_i), \psi_2(s_i')\right)}}{\sum_{j=1}^{K} e^{f\left(\psi_1(s_i, a_i), \psi_2(s_j')\right)}}\right) \tag{8}$$

$$\mathcal{L}(\psi_1, \phi_1, \phi_2) = -\sum_{i=1}^{|\mathcal{B}|} \log\left(\frac{e^{f(\phi_1(\psi_1(s_i, a_i)), \phi_2(g_i))}}{\sum_{j=1}^{K} e^{f(\phi_1(\psi_1(s_i, a_i)), \phi_2(g_j))}}\right) \tag{9}$$

$f$ is the distance function, which could be norm distance, L2 distance, dot or cosine similarity. $\psi$ and $\phi$ is neural network encoder. $s_i$ and $a_i$ are the state and action at time t in the i-th trajectory, and $s_i'$ is the state at a future time in the i-th trajectory. $s_j'$ is the state in the j-th trajectory. And $g_i$ is the goal at a future time in the i-th trajectory, $g_j$ is the goal in the j-th trajectory. Eq. (8) learn intermediate representations by contrasting $(s, a)$ and $s'$. Eq. (9) learn target representations by contrasting $\psi_1(s, a)$ and $g$.

First, we consider the sample pairs used to contrast in Eq. (9). Although the intuitive choice would be to use the state and the goal, HCLR instead used the state–action pair $(s, a)$ and the goal $g$. The reason for this design is that the actor is trained on the distance between $(s, a)$ and $g$. Before the intermediate representations converge, the distance between $(s, a)$ and $s'$ is unreliable, and inferring the distance between $(s, a)$ and $(g)$ through $s$ would only amplify this error. We conducted an error analysis in Section 4.3.

Another question is whether the intermediate representations we obtain by Eq. (8) contain information related to the distance to the goal. This is similar to the collapse problem in normal contrastive learning: we may not necessarily find a good representation even if the input contains enough information. A straightforward solution is to allow the gradients in Eq. (9) to propagate back to the encoder in Eq. (8). This could increase the mutual information entropy $I(\psi_1(s, a), g)$, which is beneficial to get better representations in Eq. (9). We also provide the proof in Section 4.3.

Finally, we discuss the sampling strategy. In Eq. (8), the sampling strategy is basically the same as that of CRL. Briefly, sample $(s_i, a)$ at random timestep $t$ and future state $s_i'$ at future timestep $t'$ in trajectory $i$ as positive pairs, and sample $s_j'$ from different trajectories in the same batch as negative pairs. The sampling strategy in Eq. (9) can be adjusted according to different goal settings in the environment. If the representations of the goals in a task is variable, then we still use the sampling strategy described above. If the representation of the goals in a task is fixed, we select $(s, a)$ from a timestep closer to the target as positive pairs, and those from an earlier timestep or other trajectories as negative pairs.

## 4.3 ANALYSIS AND PROOF

The first question is why we contrast $\psi_1(s, a)$ and $g$ in Eq. (9) instead of $\psi_2(s)$ and $g$. Firstly, $\psi_1(s, a)$ and $\psi_2(s)$ is in the same vector space. So it is basically equivalent when learning representations whether we use either one. But when we use representations in the actor of RL, we need to compare $(s, a)$ and $g$. Therefore, if we contrast $s$ and $g$, the error between $\psi_1(s, a)$ and $\psi_2(s)$ in the intermediate representation will be passed to the actor. In Section 5.4, we conducted relevant experiments.

Then, we analyze the error of our method compared to CRL Eysenbach et al. (2022). We use optimal scoring about mutual information entropy to represent the error of contrastive learning:

$$s_{\phi_1,\phi_2}(x_1, x_2) \sim log\frac{p(\phi_2(x_2)|\phi_1(x_1))}{p(\phi_2(x_2))} \tag{10}$$

$\psi$ and $\phi$ are the encoders, and the $x_1$ and $x_2$ are inputs. We can give the following proposition:

**Proposition 1.** Let $s_{\phi_1,\phi_2}(x_1, x_2)$ is Lipschitz continuous on input $x_1$. Assume there exists $\psi_0$ such that $s_{\phi_1,\phi_2}(x_1, x_2) = s_{\phi_1,\phi_2}(\psi_0(x_1), x_2)$.

If there exists $\psi_1$ satisfying $||\psi_1(x) - \psi_0(x)|| < \epsilon, \forall x \in \mathcal{A}$, then it follows that:

$$||s_{\phi_1,\phi_2}(\psi_1(x_1), x_2) - s_{\phi_1,\phi_2}(x_1, x_2)|| < L\epsilon \tag{11}$$

The proof is in Section A.1. In our method, $x_1 = (s, a), x_2 = g$. This proposition means that our method does not introduce much additional error compared to the original method under certain conditions. The assumption exists $\psi_0$ is natural as the discuss in beginning of this section. A more direct statement is: we contrast $(s, a)$ and $s$ to obtain a vector space that retains all the information needed to calculate the distance between $(s, a)$ and $g$. And $\psi_0(s, a)$ is in this vector space.

But the condition that $||\psi_1(x) - \psi_0(x)|| < \epsilon, \forall x \in \mathcal{A}$ is not natural. In Section 4.1, we give the solution: let the gradient of $\psi_1$ descend, the mutual information entropy $I(\psi_1(s, a), g)$ will increase. This solution is equivalent to satisfying the above condition. Because if the mutual information entropy is high, $\psi_1(s, a)$ is enough to be trained with $g$ for target representation, which means that $\psi_1$ is an ideal encoder.

We denote $U = \psi_1(S, A), V = \phi_1(U)$. In Section A.2, we prove the follow proposition:

**Proposition 2.** Assume $||\nabla_{\psi_1} I(U; G \mid V)|| < \epsilon$. If $||\nabla_{\psi_1} I(V; G)|| > \epsilon$, then:

$$\langle -\nabla_{\psi_1}\mathcal{L}_V, \nabla_{\psi_1} I(U; G)\rangle > 0 \tag{12}$$

Their inner product is greater than 0, which means that when we let the gradient of $\psi_1$ descend, we also increase the mutual information entropy of $V$ and $G$. This is significant to our method.

The assumption $||\nabla_{\psi_1} I(U; G \mid V)|| < \epsilon$ is easy to satisfy, in which $\epsilon$ is a small constant. It means the uncertainty between $U$ and $G$ will not be too large when $V$ is certain, which requires that $\phi_1$ is a weak compression encoder. This is consistent with RL setting. And the condition $||\nabla_{\psi_1} I(V; G)|| > \epsilon$ means that the gradient when we try to make the final representation better should be greater than a small constant. This is also a natural condition.

## 4.4 ACTOR

As in CRL (Eysenbach et al., 2022), we use the distance between learned representations as the Q-function, and the policy is trained to select actions to minimize the distance between state–action pairs and the corresponding goals. The actor loss is:

$$\mathcal{L}_\pi(\theta) = -\mathbb{E}_{p(s,a)p(g|s,a)\pi_\theta(a'|s,g)}\left[f\left(\phi_1(\psi_1(s, a')), \phi_2(g)\right)\right] \tag{13}$$

The complete algorithm is shown in Section B.

## 5 EXPERIMENTS

Our implementation is based on the open-source repository JaxGCRL (Bortkiewicz et al., 2024), which implemented a high-performance framework based on JAX and supported various Goal-conditioned environments. The experimental setup is in the Section C. We train 10M steps for each environments and run experiments on Nvidia A5000 GPU, and report the average of five random seeds.

The main objectives of experiments are: **1)** to compare the performance between HCRL and other methods; **2)** to show how we select the best parameters; **3)** to validate the quality of anintermediate representation ; **4)** to validate our designs through ablation experiments.

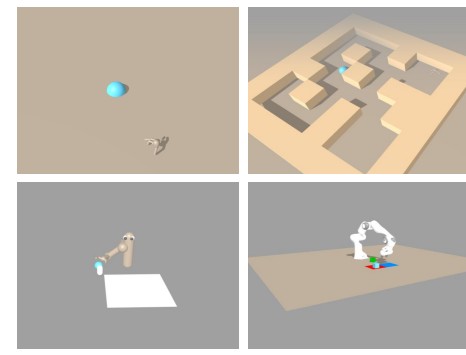

Figure 3: Some of the environments in our experiments.

### 5.1 COMPARISON EXPERIMENTS

We compare our method in JAXGCRL environments with the following approaches: Contrastive RL(CRL) (Eysenbach et al., 2022), Soft Actor-Critic (SAC) (Haarnoja et al., 2018), SAC with Hindsight Experience Replay (HER) (Andrychowicz et al., 2017), TD3 (Fujimoto et al., 2018), TD3+HER.

The parameters are followed by the recommended parameters in JAXGCRL. We selected eight complex environments from the JACGCRL and show the results in Figure 4. In these environments, SAC and TD3 find it difficult to achieve goals. HER could improve the performance of SAC and TD3. CRL performs better than SAC and TD3. Our method HCRL performs best in most environments.

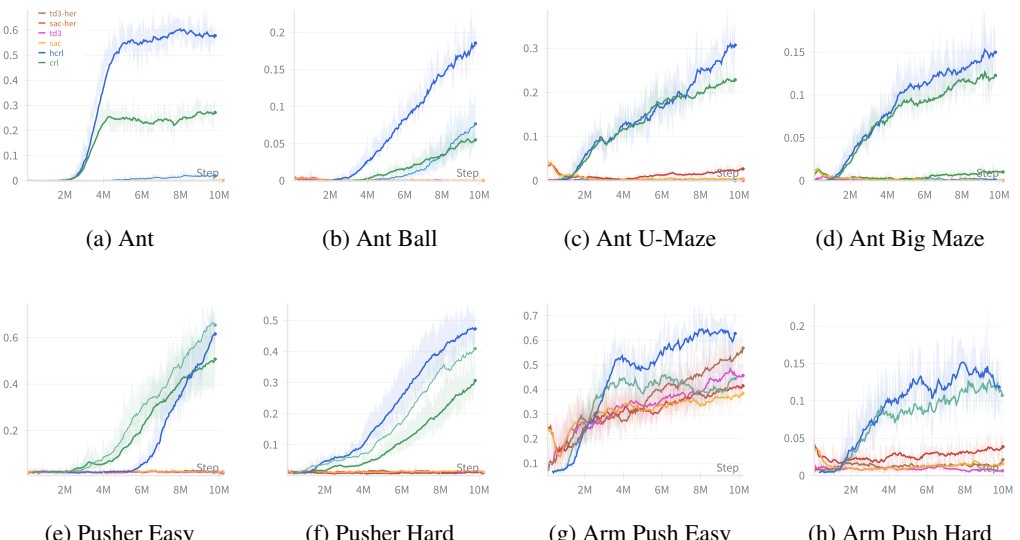

| (a) Ant | (b) Ant Ball | (c) Ant U-Maze | (d) Ant Big Maze |
| (e) Pusher Easy | (f) Pusher Hard | (g) Arm Push Easy | (h) Arm Push Hard |

Figure 4: **Success Rate of Comparison experiments.** The results show that HCRL achieved faster convergence and higher final success rates in most environments.

### 5.2 REPRESENTATION DIMENSIONS

We explored the impact of the dimensions of the intermediate representation on the results. The recommended target representation dimension is 64 (Bortkiewicz et al., 2024). Therefore, we conducted three sets of comparative experiments to verify that the dimensions of the intermediate rep-

resentations are larger, the same, and smaller than the target representation. The results are shown in Figure 5. We can see that the best performance was achieved when the intermediate representation dimension was larger than the target representation.

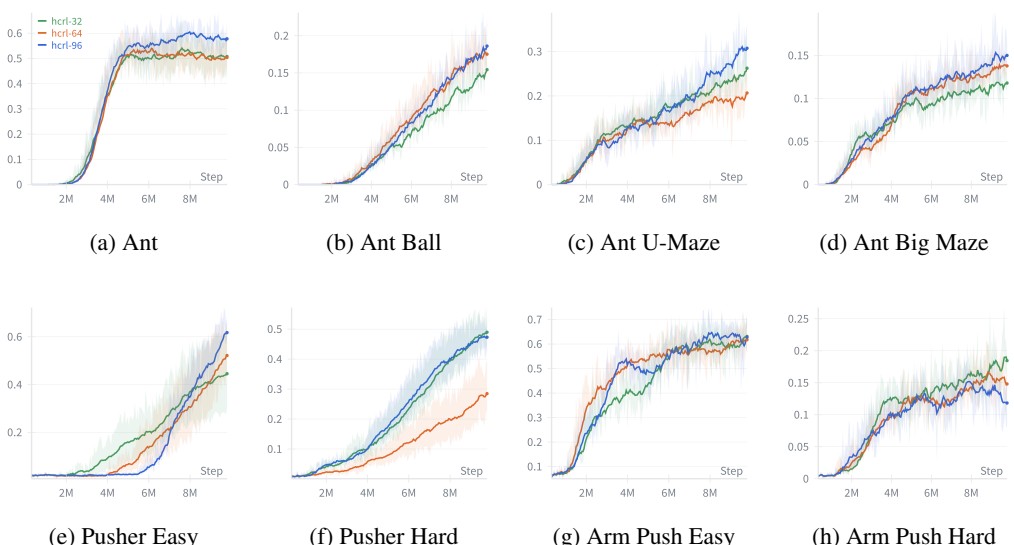

Figure 5: **Dimensions of intermediate representations.**

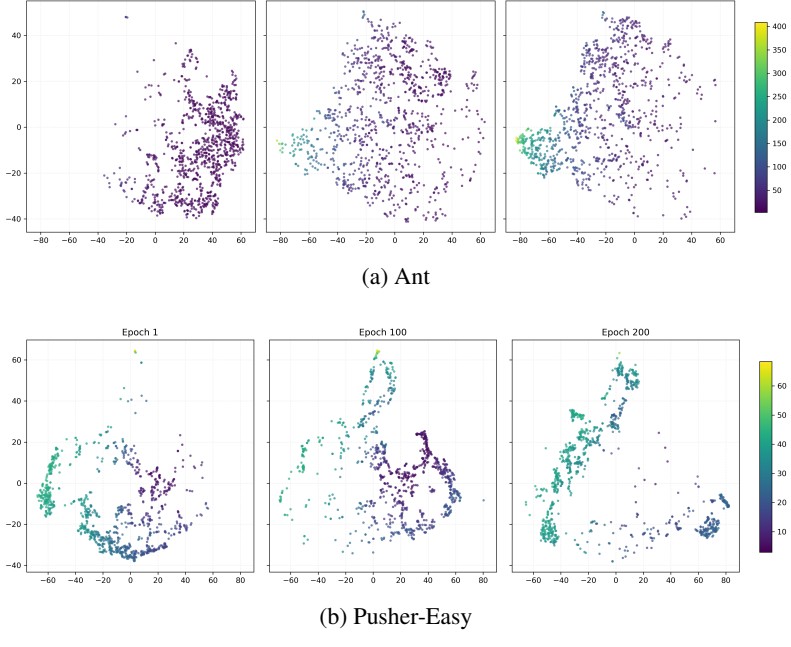

Figure 6: **Intermediate representation structure.**

### 5.3 STRUCTURED INTERMEDIATE REPRESENTATION

In CRL, the success rate directly reflects the quality of the representation. But in HCRL, we also want to know the quality of anintermediate representation. To explore this, we sampled three sets of data from the early, middle, and final stages of training. Each set of data randomly samples the current state-action and future state from the trajectory.

We expect the intermediate representations to become increasingly discriminative during training, enabling the actor to judge whether an action's resulting future state moves it closer to the goal. We used the T-SNE method to reduce the dimensionality of the state-action's intermediate representation and draw a scatter plot. The results are shown in Figure 6.

The color of the dot represents the distance from the state-action to the future state. As training progresses, the distance difference between representations gradually becomes obvious, which means that our representation does indeed structurally represent the state-action and state in the same space. Further more, the arm environment has two stages: approaching the target and moving the target, so it has a more obvious classification structure than the maze environment.

### 5.4 ABLATION EXPERIMENTS

To validate the effectiveness of our design, we conduct ablation studies by removing or modifying components of our framework. In Section 4.3, we mentioned two details: training target representations by contrasting state-action and goal instead of state and goal, allowing gradients to propagate back to the encoder $\psi_1$. We change these two designs and call them aba-1(contrast state and goal) and aba-2(stop gradient), and compare them with the original HCRL. The results are shown in Figure 7. The best performance is obtained with the original HCRL.

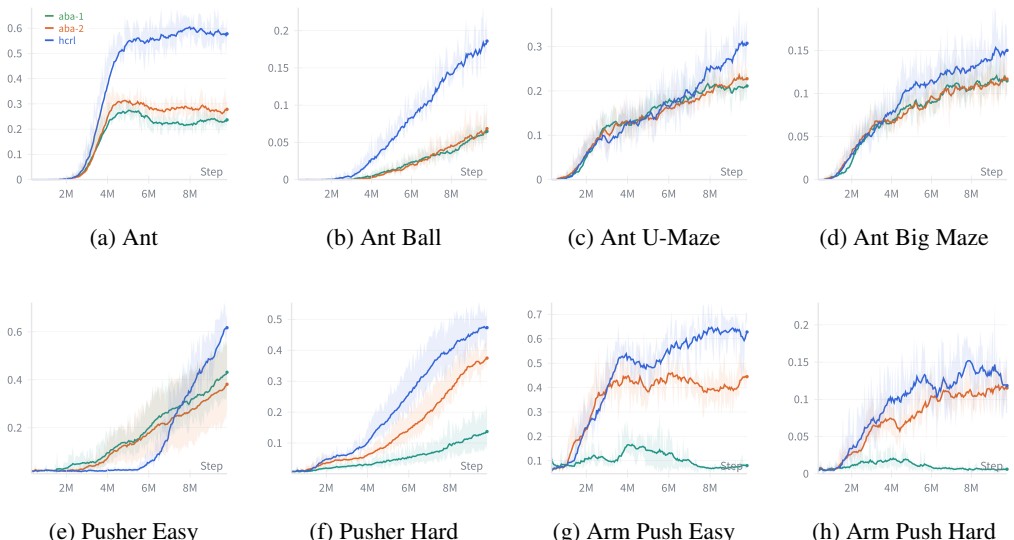

Figure 7: **Ablation experiments results.**

## 6 CONCLUSION

In this work, we show a new algorithm HCRL that could learn representations more effectively in real-world RL settings. HCRL seeks an intermediate representation related to the states before learning the representation of goals. The hierarchical structure suggests that the agent learns both the changes of states caused by actions and the distance between the states and the goals. This approach retains the advantage of CRL (Eysenbach et al., 2022), which allows contrastive learning to be directly applied to reinforcement learning without requiring reward shaping, and also improves performance and broadens its applicability to goal-conditioned RL setting. We conducted experiments on the JAXGCRL benchmark. The results show that HCRL outperforms prior methods in a range of tasks. We hope this work provides new perspectives for representation learning in RL.

**Limitations.** Based on our analysis, HCRL could also perform well when the goal involves abstract concepts, such as language instructions. However, due to limitations of simulation benchmarks and the workload required to transfer the method to real-world settings, we were unable to experimentally demonstrate this. Addressing this limitation will be the focus of our future work.

## REPRODUCIBILITY STATEMENT

We ensure that all experimental results reported in the main text and appendix are fully reproducible. The implementation of the algorithms, along with detailed instructions for running the experiments, is provided in the anonymous code repository at `https://anonymous.4open.science/r/ HCRL-6E88`. The repository allows direct usage to reproduce all reported results.

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

# A PROOFS

## A.1 PROPOSITION 1

Considering $s_{\phi_1,\phi_2}(x_1, x_2)$ is Lipschitz continuous, we have:

$$||s_{\phi_1,\phi_2}(\psi_1(x_1), x_2) - s_{\phi_1,\phi_2}(\psi_0(x_1), x_2)|| \leq L||\psi_1(x_1) - \psi_1(x_0)||$$
$$\leq L\epsilon$$

For $s_{\phi_1,\phi_2}(x_1, x_2) = s_{\phi_1,\phi_2}(\psi_0(x_1), x_2)$, we get:

$$||s_{\phi_1,\phi_2}(\psi_1(x_1), x_2) - s_{\phi_1,\phi_2}(x_1, x_2)|| < L\epsilon$$

## A.2 PROPOSITION 2

To prove this, we first introduce the following lemma:

**Lemma 1.** The mutual information and the loss of contrastive learning using InfoNCE satisfy:

$$I(V; G) \approx C - \kappa \mathcal{L}_V, \kappa > 0$$

From this, we have:

$$\nabla_{\psi_1} I(V; G) \approx -\kappa \nabla_{\psi_1} \mathcal{L}_V \qquad (14)$$

In addition, we introduce another lemma:

**Lemma 2.** If $V = \phi_1(U)$ and $\phi_1$ is a deterministic function, then:

$$I(U; G) = I(V; G) + I(U; G|V)$$

From this, we have:

$$\nabla_{\psi_1} I(U; G) = \nabla_{\psi_1} I(V; G) + \nabla_{\psi_1} I(U; G|V) \qquad (15)$$

By Eq. (14) and Eq. (15), we have:

$$\langle -\nabla_{\psi_1} \mathcal{L}_V, \nabla_{\psi_1} I(U; G) \rangle \approx \frac{1}{\kappa} \langle \nabla_{\psi_1} I(V; G), \nabla_{\psi_1} I(V; G) + \nabla_{\psi_1} I(U; G|V) \rangle$$
$$\geq \frac{1}{\kappa} \langle ||\nabla_{\psi_1} I(V; G)||^2 - ||\nabla_{\psi_1} I(V; G)|| \, ||\nabla_{\psi_1} I(U; G|V)|| \rangle$$
$$\geq \frac{1}{\kappa} \langle ||\nabla_{\psi_1} I(V; G)||(||\nabla_{\psi_1} I(V; G)|| - \epsilon) \rangle$$

If $||\nabla_{\psi_1} I(V; G)|| > \epsilon$, $\langle -\nabla_{\psi_1} \mathcal{L}_V, \nabla_{\psi_1} I(U; G) \rangle > 0$. The proposition is proved.

# B    PSEUDO CODE

---

**Algorithm 1** HCRL

---

```python
def intermediate_repr_loss(states, actions, future_states):
    sa_repr = encoder_psi_1(states, actions)
    s_repr  = encoder_psi_2(future_states)
    logits  = einsum('ik,jk->ij', sa_repr, s_repr)
    return sigmoid_binary_cross_entropy(logits=logits, labels=eye(
        batch_size))

def target_repr_loss(states, actions, goals):
    sa_repr = encoder_phi_1(encoder_psi_1(states, actions))
    g_repr  = encoder_phi_2(goals)
    logits  = einsum('ik,jk->ij', sa_repr, g_repr)
    return sigmoid_binary_cross_entropy(logits=logits, labels=eye(
        batch_size))

def actor_loss(states, goals):
    actions = policy_pi(states, goal=goals)
    sa_repr = encoder_phi_1(encoder_psi_1(states, actions))
    g_repr  = encoder_phi_2(goals)
    logits  = einsum('ik,ik->i', sa_repr, g_repr)
    return -1.0 * logits

def main():
    critic_params = [encoder_psi_1, encoder_psi_1, encoder_phi_1,
        encoder_phi_2]
    actor_params = [policy_pi]

    while(steps < total_steps):
        states, actions, future_states, achieved_goals = sample()

        loss_1 = intermediate_repr_loss(states, actions, future_states)
        critic_params.apply_gradients(loss_1)

        loss_2 = target_repr_loss(states, actions, achieved_goals)
        critic_params.apply_gradients(loss_2)

        actor_loss = actor_loss(states, achieved_goals)
        actor_params.apply_gradients(actor_loss)
```

---

# C    EXPERIMENTS SETUP

The environment of JaxGCRL  (Bortkiewicz et al., 2024) involved in the experiment is as follows:

**Ant.**  A quadruped robot in MuJoCo, required to walk to a goal sampled uniformly from a circle centered at its start position.

**Ant Ball.**  The Ant must push a movable sphere into a goal, both goal and sphere positions being randomized around the start.

**Ant U-Maze.** The Ant is placed in a "U"-shaped maze and must navigate to the target location.

**Ant Big Maze.** Similar to Ant U-Maze but with more complex maze.

**Pusher Easy.** A 3D robotic arm must push a movable object on the ground into a randomly positioned goal, with both object and goal randomized at reset.

**Pusher Hard.** Similar to Pusher Easy but with further goals.

**Arm Push Easy.** Move a cube from a random location on the blue region to a random goal on the adjacent red region. It is complex but dense-reward.

**Arm Push Hard.** Similar to Arm Push Hard but with further goals.

The hyperparameters in HCRL are shown in Table 1.

Table 1: Hyperparameters in HCRL

| hyperparameters | value |
| --- | --- |
| max_replay_size | 10000 |
| min_replay_size | 1000 |
| episode_length | 1000 |
| discounting | 0.99 |
| num_envs | 256 |
| batch_size | 256 |
| action_repeat | 1 |
| unroll_length | 62 |
| policy_lr | 3e-4 |
| critic_lr | 3e-4 |
| actor_lr | 3e-4 |
| hidden_dim | 256 |
| n_hidden | 2 |
| logsumexp_penalty | 0.1 |
| contrastive_loss_function | InfoNCE |
| representation_dimension | 64 |

## D ADDITIONAL EXPERIMENTS

### D.1 ENERGY FUNCTION

In contrastive learning, the energy function (energy_fn) measures the compatibility or similarity between sample pairs, assigning lower energy to positive pairs and higher energy to negative pairs. Prior work (Bortkiewicz et al., 2024) found contrastive RL is sensitive to energy function, which includes $L_1$ distance(norm), $L_2$ distance(l2), dot product(dot), and cosine similarity(cos).

Therefore, we conduct experiments to select the most suitable energy function. Considering the cosine similarity has a poor performance as energy function, we did not test its performance. The results are shown in Figure 8. We can see: norm energy function perform well in ant environments, l2 energy function perform well in pusher environments, and dot energy function perform well in arm environments.

We select a suitable energy function for each environment in our experiments.

### D.2 OTHER ENVIRONMENTS

Here are some results for other environments. These environments are so simple that CRL is enough, and HCRL didn't perform better. Therefore we put them into the appendix.

## E USAGE OF LLM

During the writing of this article, we used ChatGPT-5 only for text writing and grammatical polishing to improve the article's language expression and conclusions. All research design, data analysis, experimental results, and conclusions in this article were independently completed by the authors. ChatGPT-5 was not involved in any scientific judgment, data processing, or result analysis.

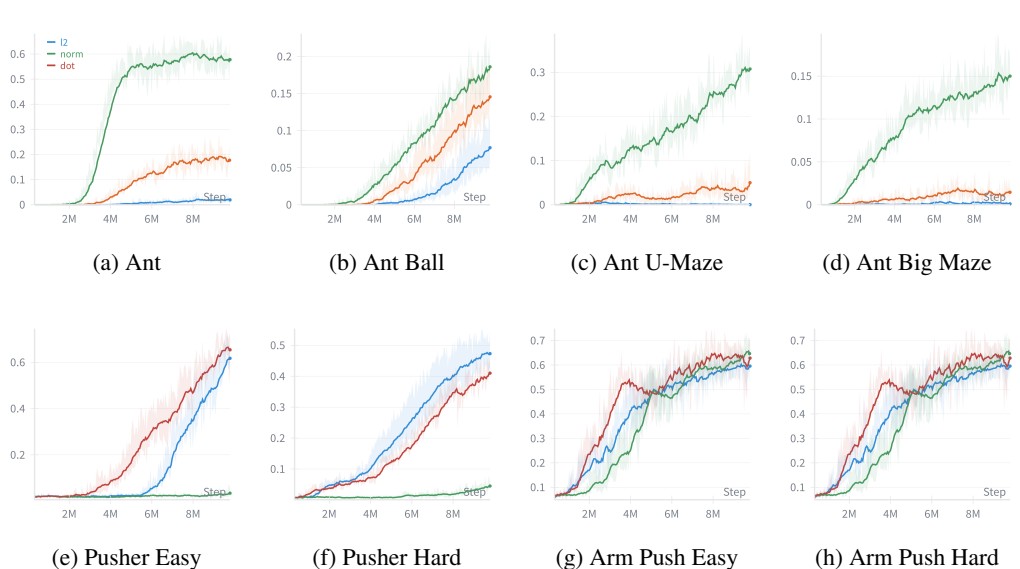

Figure 8: **Energy function for HCRL.**

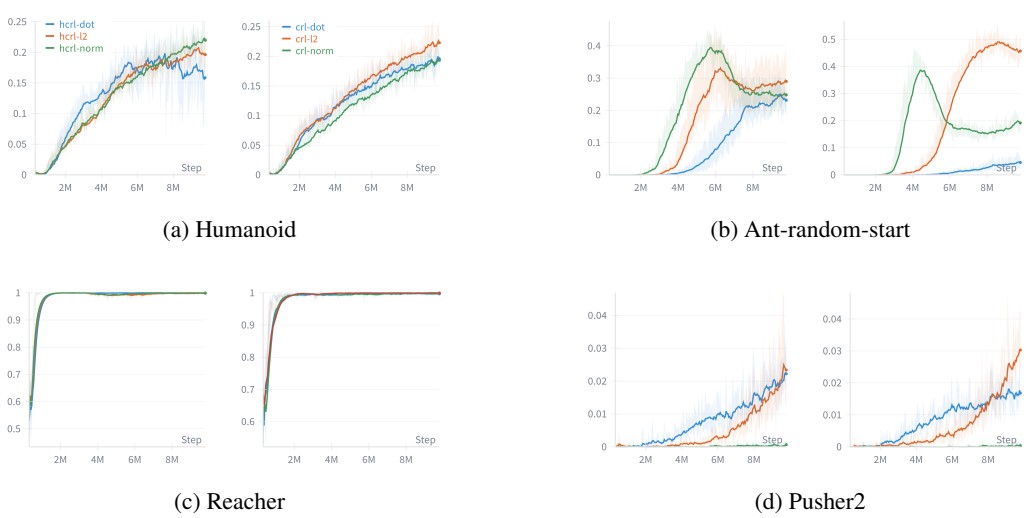

Figure 9: **Additional Environments for HCRL and CRL.**

