# OpenReview forum: "Hierarchical Contrastive Reinforcement Learning: learn representation more suitable for RL environments"
_ICLR.cc/2026/Conference — ICLR 2026 Conference Withdrawn Submission_

### Official Review · Reviewer_nF5P · 2025-10-27

**Soundness:** 2
**Presentation:** 1
**Contribution:** 2
**Rating:** 2
**Confidence:** 3

**Summary:**

This paper presents an incremental method for learning goal-conditioned reinforcement learning (GCRL) policies. The authors build on contrastive reinforcement learning and propose learning intermediate sensorimotor action-state representations using contrastive objectives. The experiments show improved results across several environments compared to a set of baselines.

The method appears is novel, albeit incremental over CRL. However, the paper suffers from grammatical issues, missing words, and a general lack of clarity. The motivation behind the proposed approach could be more clearly framed. In addition, the experiments do not demonstrate the benefits of the approach in complex state and goal spaces.

Writing and Presentation Issues

- Line 48: The meaning of “a representation that implies a probability” is unclear.
- Line 77: It is unclear what is meant.
- Line 83: Sentence appears incomplete.
- Line 134: The subtitle is awkwardly phrased.
- Abstract: The sentence “Our work fully utilizes the information in the GCRL setting” is vague. What specific information is being referred to? Moreover, the abbreviation GCRL is not introduced before use.
- Line 232: It seems the authors intended to refer to s′ (next state).
- Figures 1–3: These figures are never cited or discussed in the text.
- Lines 256–259: The intuition behind the method is unclear and should be elaborated.
- Line 285: A parenthesis is missing.
- Line 316: The subsection title “Apply Representation to RL” is unclear—consider rephrasing (e.g., “Applying Learned Representations to RL”).
- Line 377: The statement “The recommended target representation dimension is 64” requires either a citation or an experimental justification.
- Section 5.3: The qualitative results are not analyzed or discussed.
- More detail should be provided about the state and action spaces used in experiments.

Experimental Concerns

- It is unclear why the experiments in Figure 9 are not presented alongside those in Figure 4. The authors explain that these environments are “too simple to converge quickly,” which is confusing: simpler environments are typically expected to converge more easily.

- A central objective of the method is to learn intermediate representations. Therefore, experiments in more challenging settings (e.g., with image-based state and goal inputs) would strengthen the paper.

- The paper does not clearly identify which factors contribute to the method’s performance. In the ablation study, the authors should analyze success rates with and without each of the two loss functions. Without the first loss function (Equation 11), the method closely resembles CRL, differing mainly in neural network architecture. It is therefore important to clarify whether the performance improvements stem from architectural changes or from the first contrastive loss. The loss functions in Equation 11 should have distinct names for clarity.

**Strengths:**

- Interesting topics and relevant benchmark
- Preliminary experiments demonstrate significant improvements over the baselines

**Weaknesses:**

- General lack of clarity
- Incremental method, relative to CRL
- Critical experiments are missing to validate the method

**Questions:**

Please, see above.

---

> ### Author Response · Authors · 2025-11-14
>
> Thank you very much for your careful reading and suggestions. I will discuss this with you in three points: 1. What I understand and am preparing to revise immediately; 2. What I think may need revision, but I'm unsure of your true meaning; 3.further explain my motivation and discuss; 4. response you other question.
>
> **1. revise immediately**
>
> a. I have fixed most of writing and presentation issues. I apologize for the poor reading experience caused by the writing issues.
>
>
> **2. revise after communication**
>
> a. You pointed out that you hope my ablation experiments can better analyze whether the performance improvement is due to the network architecture or the loss function. But the network architecture and the loss function is closely related. The algorithm would be destroyed if i change the loss function or network architecture. I could provide this experiment result if you need, but i think it is not meaningful. Maybe you have some other suggestions? Looking forward to your reply！
>
> b. My ablation experiment removed several key components for comparison without causing the algorithm to collapse. I would very much like to discuss with you whether the existing ablation experiment setup is reasonable.
>
> **3. further explain and discuss**
>
> If my algorithm was just a simple improvement on the CLR that slightly increases the success rate, I would gladly accept your rating. However, I believe my method has more valuable aspects. Perhaps due to writing issues, I haven't clearly articulated the key points, and I would like to discuss this further with you here.
>
> One of the biggest challenges in applying reinforcement learning in real-world scenarios is the setting of the reward. Therefore, much work has discussed the issue of sparse rewards, or reward shaping. CRL provides a seemingly elegant paradigm through contrastive learning: it achieves dense and reasonable rewards without requiring special reward settings. However, when CRL is applied to real-world scenarios, there are still subtle but important issues.
>
> In goal-conditioned RL simulations, the goal is often set to be equivalent to the state, or to be a few dimensions of the state. While this simplifies the scenario, is it truly reasonable? In real life, consider placing a glass of water on a table: the state is naturally what the eye sees. But can the goal be represented by a part of the state? I believe this is unreasonable. Therefore, I restructured the algorithm to handle scenarios where the goal and state reside in different representation spaces.
>
> I won't give too many details about my algorithm, to avoid making the text too long and difficult for you to read. If you find my motivation meaningful, you could read the response to the review br7U.Or you could ask me any related questions; I'd be happy to discuss them with you!
>
>
> **4. response other question**
>
> * Figure 9 is placed separately in the appendix due to page limitations, and also because my method did not demonstrate a significant advantage in these simple environments (but it did not become worse than CRL).
> * The point is, if environments are too simple, CRL is enough to perform well. This is like in the simplest environment, DQN is enough; we don't need PPO or SAC.  My previous statement was incorrect. Sorry.
>
>
> **Finally**, I apologize again for my writing and expression issues. I have revised part of them during these days. Furthermore, I very much hope to discuss the motivation behind my algorithm with you further. I hope you can see that it is not just an incremental algorithm that can only slightly improve the success rate. I will be happy to answer any questions you may have.
>
>
> **Updated on November 20th**
>
> I've updated the PDF and fixed most of the issues.

---

> > ### Comment · Reviewer_nF5P · 2025-11-20
> >
> > I thank the authors for their answer.
> >
> > 1. I still observe writing issues, including problems that I cited in my original review. The revision introduces additional typos.
> > 2. It is possible to add phi_1 to CLR or introduce weight hyper-parameters to the loss functions to study their relative importance.ate
> > 3. "But can the goal be represented by a part of the state? I believe this is unreasonable." This is an overstatement, this is reasonable in many application scenarios.
> > " I restructured the algorithm to handle scenarios where the goal and state reside in different representation spaces." It is unclear why CRL can not handle this scenario as well.
> >
> > Overall, this work still needs substantial revisions and I keep my score.

---

> ### Author Response · Authors · 2025-11-21
>
> Sorry, maybe you read a worry version PDF? I couldn't find the sentences you quoted in my new version.
>
> And I assure I have corrected the most of important errors you originally mentioned except the issue about s'(now i fixed it). I wasn't sure if it was a problem with the website, so I re-uploaded a version. I would update the final version after addition experiments to ensure no writing problem.
>
> Besides, "add phi_1 to CLR or introduce weight hyper-parameters to the loss functions". Maybe you mean CRL instead of CLR? But I'm confused how to add phi_1 because it seems there's no place in CRL to add an additional encoder.
>
> The important question I'd like to consult you is why you believe it's reasonable to represent the goal as part of the state in the real world, especially in robotic arm or navigation tasks? For example, in a robotic arm task of moving an object, the state is of course the image, but the goal should naturally be the object's coordinates or position. I think it's reasonable for them to express in different spaces.
>
> And you mentioned that I didn't explain why CRL was inappropriate. In fact, I explained the reason in detail in the blue highlighted part of the introduction in the new version. The key point is the diversity of sampled goals.

---

> > ### Comment · Reviewer_nF5P · 2025-11-26
> > **Response to authors**
> >
> > I received a notification about a, now deleted, comment, in which the authors call me an "academic layman". Similar to R519Y, I'm sorry that the authors feel this way or are disappointed about the review, but these insults are very unprofessional.
> >
> > I would like to remind the authors that each reviewer is assigned up to 6 reviews, for this conference only, and that we freely spend a lot of time to fairly assess each contribution. We, again, spend our free time during the rebuttal period to re-assess our original review.
> >
> > The truth is that the writing of the paper does not meet ICLR scientific standards and that more analysis is needed. Regarding phi_1, it would be similar to extending psi_1 with phi_1 (as in equation 9).
> >
> >  In the (deleted) comment, the authors state that "[I] quoted sentences that weren't in my newly submitted article,
> > claiming my expression was flawed." I quoted parts of authors' response, which are still visible to everyone. Responding to reviews is a difficult exercise, but being offensive to reviewers is the wrong way to go.

---

> > > ### Author Response · Authors · 2025-11-26
> > >
> > > In addition, I'd like to know why you believe it's meaningless when goals and states are not in the same space. However, I'm prepared to withdraw the paper; you can choose not to reply.

---

> ### Author Response · Authors · 2025-11-26
>
> I apologize, as my response was AI-generated; I hoped the AI ​​would respond to an irresponsible reviewer. However, the language was too strong, so I deleted it. I only think you weren't thorough in your review process, which is absolutely irrelated to academic layman.
>
> In fact, 519Y's response angered me because it was basically nonsense, while your response only frustrated me somewhat. You pointed out many grammatical problems that I did have, which is helpful to me, but seemed completely unconcerned about my method.
>
> I accept your rating and acknowledge the shortcomings of my paper, and I am prepared to withdraw it. However, I maintain my stance of protest; I still believe you did not demonstrate a serious reviewing attitude. There was no discussion or exchange, and no focus on the method itself. For example. you mentioned my section title "RL with representations" should be adjusted. I think this is meaningless and waste both of our time.
>
> Regarding your point about the heavy workload for each reviewer, I agree that this is a difficult aspect of the current situation. However, if every reviewer gives irresponsible advice like the first response of 519Y for this reason, then it would be better to prohibit those who are unwilling to review from submitting their work. This is absolutely not my fault, nor does it concern me at all. I don't understand why this is being brought up.
>
> In summary, I have some dissatisfaction with your review, which is largely irrelevant to my methodology. However, I sincerely apologize for the accusation of academic liar in the AI-generated reply; this was not my intention, nor is it a responsibility you should bear. I deleted it immediately within minutes of posting it.

---

### Official Review · Reviewer_519Y · 2025-10-29

**Soundness:** 2
**Presentation:** 1
**Contribution:** 1
**Rating:** 2
**Confidence:** 3

**Summary:**

This paper proposes a method for learning decoupled state-to-state and state-to-goal representations for goal-conditioned reinforcement learning. While the idea is interesting and potentially valuable, the paper suffers from very poor presentation, missing key baseline comparisons, unclear notation, and numerous writing errors. Several claims are unsupported, and the experimental results lack statistical analysis. Overall, the work needs substantial rewriting and stronger empirical validation before it can be considered for acceptance.

**Strengths:**

The paper presents an interesting idea for learning decoupled state-to-state and state-to-goal representations for goal-conditioned reinforcement learning (RL). This is a promising perspective and is shown to perform better than the baselines.

**Weaknesses:**

**Presentation**
- The paper contains numerous typos, incomplete sentences, and inconsistent or incorrect notation.
- In addition, several symbols are used before being introduced (or never introduced at all). For example, $p_g$ used in equation (2) and $B$ used in equation (6) are never defined, $\phi$ used in equation (3) is introduced only later. Overall, these issues make the paper extremely confusing to read.

**Missing baselines**
- Several important baselines are missing from the comparisons, such as CURL: Contrastive Unsupervised Representations for Reinforcement Learning, HIQL: Offline Goal-Conditioned RL with Latent States as Actions, Contrastive Learning as Goal-Conditioned Reinforcement Learning, and Offline Goal-Conditioned Reinforcement Learning via f-Advantage Regression, among others.
- The paper ignores these baselines entirely and only compares against the simplest one, CRL. Even if the proposed approach does not outperform these stronger baselines, their inclusion would still be valuable, at least in the appendix.

**Unsupported claims**
- The paper also includes unsupported statements such as “This setting is beneficial to enhance the generalization of RL and is meaningful to sim2real,” and “In some papers, this sparse reward function is equal to…”. Such claims should be substantiated with references or evidence.
- Finally, the authors do not report the statistical significance of their results. There are no standard deviation plots, min–max regions, or p-value analyses, which raises concerns about the robustness of the reported performance.

**Questions:**

- Why did the authors choose the baselines they did, and why not include more relevant ones?

---

> ### Author Response · Authors · 2025-11-14
>
> 1.I have tried my best to fix writing and presentation issues, including sentences and notation, and uploaded new version. I'm very sorry for the poor reading experience.
>
>  **2. I want to have a serious discussion with you about your harsh criticism of my baseline. You mentioned I did not compare my work with four key algorithms: CURL, HIQL, CLGCRL, GoFar.**
>
> * But my baseline CRL is just short name for CLGCRL. And it is also referred to in many other papers. And I point this in my paper.
>
> * Besides, the CRL paper has provided a detailed comparison to demonstrate that its performance is superior to CURL.
>
> * And HIQL and GoFar is offline RL algorithms, which need offline dataset to train. My algorithm is online algorithm, so i think it is meaningless to compare to them.
>
>
> In addition, If you're interested in what problem this algorithm actually solves, I'd be happy to discuss it with you. I've emphasized this point in discussions with other reviewers.
>
> I apologize again for my writing problems; I will try my best to revise it.
>
> **Updated on November 20th**
>
> I've updated the PDF and fixed most of the issues.

---

> > ### Comment · Reviewer_519Y · 2025-11-25
> >
> > I thank the author for their responses, but, similar to Reviewer nF5P, I believe this submission requires further revisions before it is ready for acceptance. My score remains the same.

---

> > > ### Author Response · Authors · 2025-11-26
> > >
> > > I cannot change your assessment. However, I strongly protest against you. Were you responding with an AI? You brought up some meaningless comparison algorithms and completely ignored my response. There was  no discussion or exchange. A terrible experience.

---

> > > > ### Comment · Reviewer_519Y · 2025-11-26
> > > >
> > > > I am sorry the author feels this way. None of my reviews were AI generated.
> > > >
> > > > Here are examples of continued writing issues:
> > > >
> > > > --"If the agent could achieve the goal g ∈ G, this trajectory is considered successful. According to the different reward functions r, goal-conditioned RL problems are divided into nonsparse, which reward function is similar to normal RL problems, and sparse, which reward function is generally rg = 1 if st = f(g) else 0." Should have a citation here
> > > >
> > > > --"The goal-conditioned RL algorithms try to find an optimal policy π(a|s, g). Denote pg(sg) as the distribution of acceptable termination states sg when the objective is g, and denote π(τ |sg) as the probability of sampling an infinite-length trajectory τ = (s0, a0, s1, a1, ...)..." Confusing sentence, used \pi for both policy and distribution over trajectories.
> > > >
> > > > --"First, we sample a state–action pair (s, a) from a trajectory at a given timestep and select a future state s ′ from the same trajectory, which we map into an intermediate representation space denoted ψ. Secondly, we sample a goal g from a trajectory at a given timestep and select a past state–action pair (s, a) from the same trajectory, mapping g and ψ(s, a) into the target representation space." \psi is used both for a representation space, and a function.
> > > >
> > > > --“We selected eight complex environments from the JACGCRL". JACGCRL instead of JAXGCRL.
> > > >
> > > > Other concerns about the plots:
> > > >
> > > > The shaded area in Figure 4 is unclear. For example, in Figure 4 (a) the blue curve for hcrl lies outside the shaded area for steps between 2-4M. How is this possible? what does the shaded area represent?
> > > >
> > > > Baselines:
> > > >
> > > > Even if the CRL paper compared against CURL, did they compare against the same environments you are now testing your method in? From a quick review of the CRL paper, it does not appear that the environments are the same. Therefore, I still believe it is necessary to include CURL.

---

### Official Review · Reviewer_qe3R · 2025-10-31

**Soundness:** 2
**Presentation:** 3
**Contribution:** 2
**Rating:** 2
**Confidence:** 4

**Summary:**

The authors propose HCRL, which additionally learns intermediate representations to simplify target representation learning and leverages them for policy improvement in reinforcement learning (RL). They claim this addresses the limitation of existing contrastive learning-based representation methods, which struggle to effectively capture state representations relative to goals.

**Strengths:**

Deriving intermediate representations through a hierarchical structure and applying contrastive learning across both stages appears to distinguish this work from prior studies.

**Weaknesses:**

Overall, the explanations supporting the authors' claims feel inadequate. Inferring from the paper's descriptions, the authors seem to treat goals and states as defined in separate spaces. If accurate, this constitutes a crucial assumption. Many studies, including those on CRL, presuppose that the state space and goal space are identical; thus, adopting a different setup would necessitate explicit discussion. Yet, the paper lacks any such clarification or details on how goals are defined in the experimental settings. Moreover, if baselines receive only goals with limited information while HCRL is provided with intermediate states containing additional details, this could undermine the fairness of comparisons.

**Questions:**

1. As noted in the Weakness section, is it correct that the goal space and state space are not identical?
2. In line 73, the authors assert that the encoder vector for goals is non-discriminative, whereas it is discriminative for states. Supporting evidence for this claim should be added, such as references to prior work addressing the issue or experimental demonstrations.
3. Line 287 mentions “q and k are inputs.” What exactly do q and k refer to?
4. I am curious about how goals are defined in the experimental environments employed.
5. The paper states that the experimental graphs represent averages over five random seeds. Including standard deviations across these seeds in the graphs would be advisable.
6. In line 837, further clarification is needed on how sampling is performed for states, actions, future_states, and achieved_goals.

---

> ### Author Response · Authors · 2025-11-14
>
> Thank you for your valuable comments. I will focus on discussing the details about the goal space and state space, which is related to question 1 and question 4.
>
> One of the biggest challenges in applying reinforcement learning in real-world scenarios is the setting of the reward. CRL achieves dense and reasonable rewards without requiring special reward settings. But CRL is still only suitable for simulation environments and not for real-world environments. Specifically：
>
> * Many simulation environments equate the goal to the future state, while some set the goal as part of dimension in state vector. But this is not reasonable in real world.
>
> * In most real-world scenarios, the representation of a goal and the representation of a state are completely unrelated. For example, a goal might be a language instruction like "put the apple in the plate.". This is very different to sim and CRL will not work. The HCRL architecture can still handle this scenario because it doesn't require an explicit relationship between the state and the goal. In fact, I should conduct more experiments with real robotic arms to emphasize this conclusion, but that process is quite complex. Our future work revolves around this point.
>
> * Besides, When goal is part of dimension in state vector, such as "FetchPush" env, the goal is object position and the the state is robotics arm position + object position, which is similar to some simple case in real world. And CRL still has room to improve in this kind of environment. And we provide a detailed analysis in the new version of the paper.
>
>
> In conclusion, I think CRL is a very great algorithm. But we need a new algorithm to solve the difference about goal space and state space in real world.
>
>
> **Response to other questions**
>
> * How to set goal in our env? In my experiments, the goal of most environments is part of state and HCRL could perform better. In addition, HCRL could work well when the goal is unrelated to the state, such as in language commands. However, this part is not yet complete and is not reflected in this paper.
>
> * In weakness, you mentioned "baselines receive only goals with limited information".  In fact, I didn't impose any restrictions on the baseline algorithm. Instead, one of the focuses of this work is to make fuller use of information from the environment when applying contrastive learning to reinforcement learning. If the information is there, but the baseline algorithm （and other related algorithms）doesn't use it, that's obviously an area that deserves improvement.
>
> * For question 2, I think the conclusion is clear when goal space and state space are different. For example, the task is to use a robotic arm to place an apple on a table. The target (the apple) doesn't move until the robotic arm touches it. This is meaningless for contrastive learning. However, the state is constantly changing. In addition, i'm considering to conduct related experiments to prove this.
>
> * For question 3, The q and k  is my writing error. I would revise it. The correct notation would be x_1 and x_2.
>
> * For question 5, I would modify my figure in paper to add instandard deviations.
>
> * For question 6, the sample methods is similar to HER and CRL. I added the details about sampling into Section Method in papers.
>
>
> Thank you very much for your detailed comments and insightful observations on this paper. I will carefully revise the relevant issues. I also look forward to further discussions with you regarding the goal space and state space. What I want to emphasize is that this is not a random assumption, but a question worth considering regarding the transfer of reinforcement learning to real-world environments. Looking forward to your reply!
>
>
> **Updated on November 20th**
>
> I've updated the PDF and fixed most of the issues.
>
> **Updated on November 26th**
>
> We decided to withdraw the paper, so you don't need to spend time replying. Of course, I would also pleasure to hear any suggestions or discussions you may have.

---

### Official Review · Reviewer_br7U · 2025-11-05

**Soundness:** 2
**Presentation:** 3
**Contribution:** 3
**Rating:** 6
**Confidence:** 4

**Summary:**

The paper presents an interesting adaptation of contrastive reinforcement learning (CRL) for the specific GCRL setting where the goal representation differs from the full state representation. This is a practical and interesting setting affecting many real-world use cases of GCRL. The idea is to use the state trajectory to learn a useful intermediate representation of the state, and then use this representation to learn effective goal distances. The paper provides evidence of the success of this methodology on a suite of GCRL environments, demonstrating superior performance compared to CRL and other state-of-the-art goal-conditioned RL algorithms.

**Strengths:**

- The paper is supported by a comprehensive set of experiments across a diverse suite of challenging GCRL environments. The results convincingly demonstrate that HCRL significantly outperforms strong baselines, including CRL and HER-augmented agents, in terms of both sample efficiency and final success rate.

- The paper is well-written and clearly structured and the methodology is presented in a logical flow that is easy to follow.

- The paper addresses a practical and important problem in GCRL, the common scenario where the goal is a sub-component of the full state.

**Weaknesses:**

- The contribution doesn't seem to be much novel compared to the original CRL. Moreover the problem doesn't seem to be very well motivated, I think the characteristic of the goal space where this intermediate representation would help have to be improved and is not easy for me to understand this limitation and when this extra representation could hurt instead of helping learning.

- While CRL is the most direct baseline, the comparison would be more compelling with the inclusion of other representation-learning or contrastive-based GCRL methods, such as QRL (Wang et. al. Optimal Goal-Reaching Reinforcement Learning via Quasimetric Learning).

**Questions:**

- See weaknesses points.

- Why repeating equations 8 and 9 again in equation 10 and 11?

- What would happen if the goal represent a set of acceptable terminal states, could the precision learned by the intermediate representation directly interfere with the necessary goal-set invariance?

---

> ### Author Response · Authors · 2025-11-14
>
> Thank you very much for your recognition of my work! Your suggestions and questions are also very important. I will revise and discuss them.
>
> **To Revise**
>
>
> **1. Thank you very much for pointing out that QRL can be used as one of the baselines**
>
> In fact, in some simulation environments, QRL does outperform CRL and HCRL. However, QRL's implementation has a clear limitation: it uses `future_state` to represent the goal, which is related to its distance design. If the goal cannot be represented using `future_state`, then QRL seems unable to function correctly.
>
> I will explain in detail what constitutes a more reasonable goal representation later.
>
> **2. I would be very pleasure to explain the motivation behind this method in detail.**
>
> Reward is one of the key issues in applying reinforcement learning to the real world. CRL provides a seemingly elegant paradigm through contrastive learning: it achieves dense and reasonable rewards without requiring special reward settings. But  CRL is still only suitable for simulation environments and not for real-world environments.
> * Many simulation environments equate the goal to the future state, while some set the goal as part of dimension in state vector. But this is not reasonable in real world.
> * In most real-world scenarios, the representation of a goal and the representation of a state are completely unrelated. For example, a goal might be a language instruction like "put the apple in the plate.". This is very different to sim and CRL  will not work. The HCRL architecture can still handle this scenario because it doesn't require an explicit relationship between the state and the goal. In fact, I should conduct more experiments with real robotic arms to emphasize this conclusion, but that process is quite complex. Our current work revolves around this point.
> * Besides, When goal is part of dimension in state vector, such as "FetchPush" env, the goal is object position and the the state is robotics arm position + object position, which is similar to some simple case in real world.  And CRL still has room to improve in this kind of environment. And we provide a detailed analysis in the new version of the paper.
>
> **3. The reason repeating equations 8 and 9 again in equation 10 and 11 is**
>
>  8 and 9 is just the loss funciton. 10 and 11 indicates where the gradient should be propagated to.  In the initial design, the loss in Equation 9 only updated the φ encoder, but the results were unsatisfactory. In the revised version, I allowed the loss in Equation 9 to be propagated to the ψ encoder.  I have revised this part in the paper to avoid confusing。
>
>
> **Response to your question**
>
> **1. If there is a set of acceptable terminal states, how would the algorithm perform?**
> * This seems to be related to multi-objective tasks but I'm not entirely sure what you mean. There are two different multi-objective environments.
> * One simple scenario is that, in the same environment, a random objective is assigned before each action is taken. Another scenario is that the objective is vaguely expressed, so one expression might correspond to multiple acceptable states, such as "put the apple on the table". For the first one, the JAXGCRL environment (the benchmark we selected) is basically  belong to this category , and our algorithm could perform well, and .
> * But for the second one, there is no benchmark specifically used for this. So we plan to verify the effectiveness in reality, and this part is also mentioned in the limitations of the updated paper. Theoretically, in this more realistic scenario, HCRL should be better than CRL.
>
>
>
> **Finally**, thank you again for your careful reading and comments on this work! Your questions are very insightful. I am working hard to revise it based on your feedback! Looking forward to more communications with you!
>
> **Updated on November 20th**
>
> I've updated the PDF and fixed most of the issues.
>
> **Updated on November 26th**
>
> We decided to withdraw the paper, so you don't need to spend time replying. Of course, I would also pleasure to hear any suggestions or discussions you may have.

---

### Note · Authors · 2025-11-26

I have read and agree with the venue's withdrawal policy on behalf of myself and my co-authors.